# Parental effects of Bt toxin and vitamin A on *Helicoverpa armigera*

**Carmen López, Daniela Zanga, Alejandro Juárez-Escario, Pilar Muñoz, Matilde Eizaguirre** *

Department of Plant Production and Forestry Sciences, University of Lleida-Agrotecnio Center, Lleida, Spain

* matilde.eizaguirre@udl.cat

**Data Availability Statement:** All relevant data are within the paper and its Supporting Information files.

**Funding:** This study received funding from Ministerio de Ciencia, Innovación y Universidades

## Abstract

The increase in the area cultivated with vitamin-enriched transgenic crops producing Bt toxin raises the question of whether the addition of vitamins will in any way mitigates the effect of the toxin on the phytophagous insects that feed on those crops. On the other hand, the parental effect that feeding on these enriched transgenic crops may have on the offspring of the phytophagous that survive on them is not well known. In this work, the effect of vitamin A (β-carotene) addition to diets with or without Bt toxin on *Helicoverpa armigera* larvae and their offspring was determined. The addition of vitamin A did not have any beneficial effect either for the larvae fed on enriched diets nor for their offspring. However, parental effects due to dietary feeding with the toxin were detected since adults from larvae fed on the Bt diet had higher mating success than those fed on the toxin-free diet, although there were no differences on the fertility of mated females regardless of whether their previous larvae fed on the Bt or non-Bt diet. A certain adaptive effect to the toxin was also noted since the mortality of larvae whose previous generation fed on diet with Bt was lower than that of the larvae that came from larvae fed on a non-Bt diet. It would be interesting to determine if *H. armigera* adults prefer to mate and lay eggs in the same type of crops in which they have developed or if feeding on different crops, such as corn or alfalfa, causes different paternal effects on the offspring. These aspects can be of great importance in the development of resistance of this species to the Bt toxin.

## Introduction

The portion of cultivated land devoted to transgenic crop production has been steadily increasing since being introduced in 1996. Since the introduction of transgenic crops, their characteristics have diversified to include some that incorporate the toxin produced by *Bacillus thuringiensis* (Bt), an entomopathogenic bacterium. In 2018, 26 countries, 21 developed and 5 industrialized, planted 191.7 million hectares of transgenic crops, most of which had two or more Bt toxins or were tolerant to herbicides [1]. Furthermore, many new biofortified conventional and transgenic crops have been developed, with improved nutritional characteristics such as high levels of vitamin A (β-carotene) or vitamin C (Ascorbic acid: AsA), and are now becoming commercialized [2, 3]. Some examples of these biofortified transgenic crops under

(AGL2017-84127-R). The funders had no role in study design, data collection and analysis, decision to publish, or preparation of the manuscript.

development are biofortified transgenic sorghum for nutritional enhancement [4] and the "golden" rice, which has been recently authorized for consumption in the Philippines [5].

The increasing growth of vitamin-biofortified crops raises questions about the potential effects on the development and mortality of phytophagous insects that feed on these enriched crops. One of these questions is whether biofortified crops favor populations of phytophagous pests. To date, these concerns have received little attention. Unsurprisingly, little is known about the effects of biofortified Bt crops on phytophagous insects, or how an increase in vitamins could affect their survival, development, and behavior. Zanga *et al.* [6] found that the addition of β-carotene to a Bt diet moderated the efficacy of the Bt toxin, reducing larval mortality in the Bt target pest, *Ostrinia nubilalis* Hübner (Lepidoptera: Crambidae). It was hypothesized that the Bt toxin was less effective in the β-carotene groups as a result of the increased activity of enzymes involved in detoxification mechanisms, such as catalase (CAT), superoxide dismutase (SOD), and glutathione S-transferase (GST). Conversely, Girón-Calva *et al.* [7] demonstrated that newborn *O. nubilalis* larvae fed Bt diets exhibited higher mortality when supplemented with β-carotene. Similar studies focusing on caterpillars that are less susceptible to Bt toxins are particularly interesting, as these organisms may still be favored by the higher vitamin content of Bt maize plants. For instance, Lopez *et al.* [8] determined that the response to a vitamin A- and C-enriched Bt diet was somewhat different between two secondary maize pests, *Helicoverpa armigera* (Hübner) and *Mythimna unipuncta* (Haworth) (Lepidoptera: Noctuidae). Ingestion of the Bt diet caused oxidative stress in both species and resulted in longer larval development and lower pupal weight. These effects were not mitigated by the presence of vitamins in the diet; however, vitamin enrichment reduced the mortality of Bt-fed *H. armigera*. In addition, β-carotene decreased the activity of GST in both species, suggesting that this enzyme has an antioxidant role. The study demonstrated that biofortified Bt crops should not enhance the development of *H. armigera* or affect the development of *M. unipuncta*. However, the effects of increased vitamins could be highly variable amongst different species and should, therefore, be studied in each specific phytophagous.

Eizaguirre *et al.* [9] elucidated that the larvae of *H. armigera*, a polyphagous [10, 11] cosmopolitan pest whose larvae have a high capacity to develop resistance to insecticides [12, 13] and Bt toxins [14, 15], could survive and complete their development even when feeding on Bt maize. The introduction of biofortified Bt cultures could thereby improve the fitness of this low-susceptibility species when it feeds on these crops.

Due to the increase in temperature and irrigated areas in the region where this study was conducted, maize has changed from being cultivated in an annual cycle to more intense alternatives [16]. These include long-cycle grain maize sown in early spring and shorter-cycle forage maize sown after the harvest of winter cereal. This change in cultivation favors the presence of ears with tender silks from early spring to late fall and allows *H. armigera* to complete several successive generations. Despite scientists having recognized the immense influence of parental nutrition on progeny phenotype [17], there are few studies assessing the mating behavior of adults and the development of the next generation of larvae when fed Bt or non-Bt crops.

Similarly, studies assessing the potential consequences of a diet containing biofortified Bt-maize on the development, mortality, and final weight of the next generation of larvae are uncommon. Studies of this nature are significant because a portion of the *H. armigera* larvae that feed on Bt transgenic maize survives and pupates. These pupae are usually smaller in size and weight than those fed a non-Bt diet [8]. Because the weight of lepidopteran pupae has been associated with the fertility of the resulting adults [18], it is conceivable that adults from heavier pupae (larvae fed non-Bt maize) will present with higher fertility than adults from lighter pupae (larvae fed Bt maize). However, under field conditions, it is expected that adults

emerging in a determinate field (Bt or non-Bt) mate in the same field and then disperse to lay eggs in neighboring fields [19] (Bt or non-Bt). Studies about whether the fact that the larvae fed on Bt or non-Bt maize enriched or not with vitamins will have consequences on development, mortality or final weight of larvae of the next generation are uncommon. Accordingly, this study aimed to determine the possible effects of parental diets, (i) with or without Bt toxin (ii) supplemented or not supplemented with vitamin A, on the fertility of the next generation of adults and the performance of the subsequent larvae fed again on a diet with Bt or non-Bt maize supplemented or not with vitamin A.

## Materials and methods

### Insect general rearing

To establish the population required for the experiments, *H. armigera* larvae were collected from alfalfa crops, with permission of the owner (Josep Piqué), in Lleida (GPS coordinates 41˚ 37'39.1"N, 0˚35'39.5"E). Research on insects does not need approval from an Institutional Animal Care and Use Committee (IACUC) [20]. All field-collected larvae (F0) were individually collected to avoid cannibalism. The general rearing conditions were 25˚C temperature and 16:8 light: dark (LD) (h) photoperiod and relative humidity ($>$ 60%). Larvae were fed until pupae emergence with the semi artificial diet (Fig 1) [21]. The pupae were maintained at 15˚C and 16:8 LD photoperiod to synchronize adult emergence. When the number of females and males was considered sufficient to full a mating cage (6 females and 5 males per cage, [8]) the pupae were placed in the general rearing conditions in pots with daily moistened soil. Adults (F0) from these pupae were placed in the mating cages (cylindrical methacrylate tubes of 25 cm diameter and 40 cm height, closed at the base with an expanded polystyrene lid, and sealed at the top with white gauze) and given a drinker of honey solution and a cotton ball for egg laying. Eggs were maintained until hatching in the same rearing conditions.

### Lyophilized maize

Leaves from Bt (DKC6667Y, with Cry1Ab, MON810) and non-Bt (DKC6666, isogenic) maize were collected from unsprayed maize crops near Lleida during the summer of 2017. The leaves were cut into small strips, and the main veins were removed. The plant material was freeze-dried in a vacuum drier (Gamma 2–16 LSC plus, CHRIST, Osterode am Harz, Germany) and

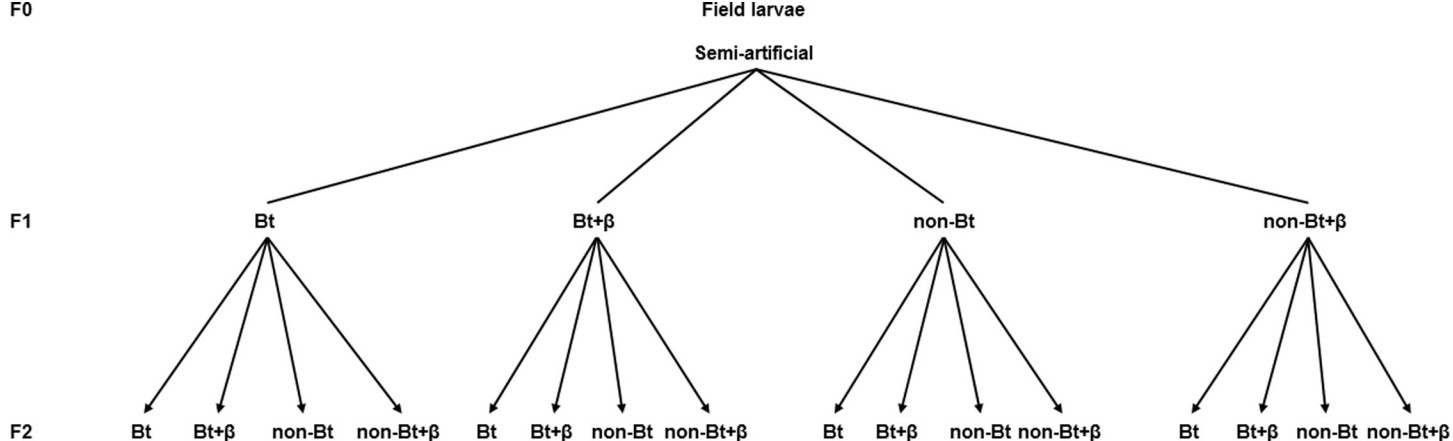

**Fig 1. Scheme of the performed experiments and the diets used for each generation.** Bt indicates diets with lyophilized leaves of Bt maize; non-Bt indicates diets with lyophilized leaves of non-Bt maize; +β indicates the addition of vitamin A (β-carotene: 0.6% in weight) to the corresponding diets.

ground in a Thermomix® until a fine powder was obtained. The lyophilized material was stored at -80˚C for later use.

## Diets

The diets used in these experiments were prepared by modifying the diet of noctuids from Eizaguirre and Albajes [21]. To obtain diets with sublethal doses of the Bt toxin, 25% of maize flour was replaced with lyophilized Bt or non-Bt maize leaves, with or without 0.6% of β-carotene. The amount of lyophilized Bt was chosen according to our previous study [22] in order to obtain a low toxin concentration in the diet (approximately 160 ng Cry 1Ab g$^{-1}$ fresh weight) in the range of commercial transgenic maize [23]. The β-carotene concentration was decided according to our previous works [6, 8]. The resulting diets (Fig 1) were: diet made with non-Bt leaves (non-Bt), non-Bt leaves with 0.6% β-carotene (non-Bt+β), Bt leaves (Bt), and Bt leaves with 0.6% β-carotene (Bt+β).

## Experiments

**Experiments with the F1 larvae and adults.** The newly hatched F1 larvae were divided into 4 groups ($\approx$ 30 larvae per group) and fed 1 of the 4 experimental diets. Larvae were individually placed in small rearing boxes (5.5 cm diameter × 3 cm high) with a small portion of the corresponding diet. Larvae were checked every 2 or 3 days, and the diet was replaced when necessary. Larval mortality, larval development duration, and pupal weight were daily recorded. The mortality due to rearing management was not included in the calculations (e.g., larvae dead before eating the diet).

The resulting F1 pupae were characterized based on sex and parental diet, and kept at 15˚C and 16:8 (LD) photoperiod to synchronize adult emergence. Once there was sufficient pupae for a mating cage, the procedure was the previously described for the F0. The emerging F1 adults were transferred to mating cages with a honey solution for feeding and a piece of hydrophilic cotton for egg laying. The mating experiment was performed using 36 females (6 cages with 6 females per /cage) and 30 males (6 cages with 5 males per /cage) for each parental diet.

Cotton pieces and drinkers were changed every two days. Cotton pieces containing eggs were identified and placed in separate boxes at 25˚C to evaluate egg evolution. The neonate larvae (F2) and fertilized eggs were counted once the first larvae emerged (4 to 5 days) and the fertilized eggs were distinguishable by the obvious presence of the head. As each cotton piece had eggs of only two consecutive days, larval emergence was synchronized to avoid sampling error (resulting from counting the many infertile eggs of unmated females). Hatched larvae were placed in rearing boxes with the appropriate diet if it was necessary to continue the experiment, if not, they were frozen.

Once the F1 adults died, they were frozen for later verification of mating status. The number of mated females (F1) and the total number of larvae (F2) obtained from the mated females were analyzed.

**Experiments with F2 larvae.** At least 53 F2 larvae were randomly selected from the different mating cages of each treatment and used for the F2 experiments. To determine the effect of the different progeny diets (in addition to the effect of different parental diets) on the F2 larval development, F2 larvae were divided into 4 groups based on parental diet and each of these 4 groups was further divided based on the 4 diets used in the previous experiment (Fig 1). As a result, we obtained 16 different groups, with combinations considering the feeding of parental and progeny. Additionally, larval mortality, larval development duration, and pupal weight were recorded.

**Statistical analysis.** To assess the effect of diet and sex on development duration and F1 pupal weight, a two-way analysis of variance (ANOVA) was used (diet * sex). To analyze the effect of each component of the diet on the same biological traits, a three-way ANOVA was used (Bt * β-carotene * sex), and a student's *t*-test was used for least squares means comparison.

Mortality was calculated as a percentage in each trial and was analyzed using a generalized linear model (GLM) with a binomial distribution. Mortality was assessed in relation to the influence of diet and dietary components (Bt and β-carotene).

Mating rate was calculated as the percentage of mated females in each mating cage and analyzed using a GLM with a binomial distribution. Fertility was calculated as the number of fertile eggs from mated females.

For the second-generation study, we added the parent factor (i.e., the diet ingested by the parents of the studied larvae). In this way, the analysis performed was a two-way (parents * diet) ANOVA on development duration and pupal weight, and a two-way ANOVA (Bt * β-carotene) for the effect of the dietary components. The effect of parental diet was also included in the mortality analyses.

All statistical tests were performed using JMP® Pro 15.2.0 [24]. In all statistical analyses, statistical significance was set at $P < 0.05$.

## Results

### F1 development and mortality

The larval development duration (in days) of *H. armigera* F1 larvae fed diets with or without Bt and with or without β-carotene is shown in Fig 2.

The duration of larval development of *H. armigera* depended on the type of diet ingested ($F_{101, 3} = 66.83$, $P = 0.0001$); however, there were no differences in the duration according to the sex of the resulting pupae ($F_{101, 1} = 0.0158$, $P = 0.9003$). Diet component analysis indicated that the presence of Bt toxin in the diet prolonged larval stage development ($F_{101, 1} = 197.25$; $P = 0.0001$). The addition of β-carotene alone did not influence this duration ($F_{101, 1} = 0.6736$; $P = 0.4138$); however, there was an interaction between Bt and β-carotene ($F_{101, 1} = 6.0852$; $P = 0.0154$) (Fig 2), in that β-carotene did not affect larval duration when they were fed Bt diets but prolonged larval development when they were fed non-Bt diets (Fig 2)

Statistical analysis of the weight (Fig 3) of the resulting F1 pupae indicated that larval diet had a significant effect on weight ($F_{101, 3} = 5.86$; $P = 0.0010$) but not on sex ($F_{101, 1} = 0.26$; $P = 0.6099$). Analysis of larval dietary components demonstrated that the presence of β-carotene affected the final pupal weight ($F_{101, 1} = 7.3867$; $P = 0.0078$). Similarly, the presence of the Bt toxin affected pupal weight ($F_{101, 1} = 10.97$; $P = 0.0013$) (Fig 3). These analyses indicate that pupae from larvae fed Bt diets weighed less than those fed non-Bt diets. Moreover, pupae from larvae fed diets with β-carotene weighed less than pupae from larvae fed diets without β-carotene.

Mortality, calculated as the percentage of dead larvae in each group, was analyzed using a GLM with a binomial distribution (Table 1). The results indicated that there were no differences in larval mortality based on diet type (Bt or non-Bt, with or without β-carotene) [Diet ($χ^2 = 3.7410$; $P_{χ2} = 0.2908$); Bt toxin ($χ^2 = 0.00296$; $P_{χ2} = 0.9566$); β-carotene ($χ^2 = 0.8531$; $P_{χ2} = 0.3557$)].

### F1 mating and fertility

The percentage of F1 mated females (Fig 4) in each trial was analyzed using a GLM with a binomial distribution. The results indicated a statistically significant effect of diet ($χ^2 = 9.14$, fd = 3, $P_{χ2} = 0.0275$). The diet component analysis indicated that the presence of the Bt toxin

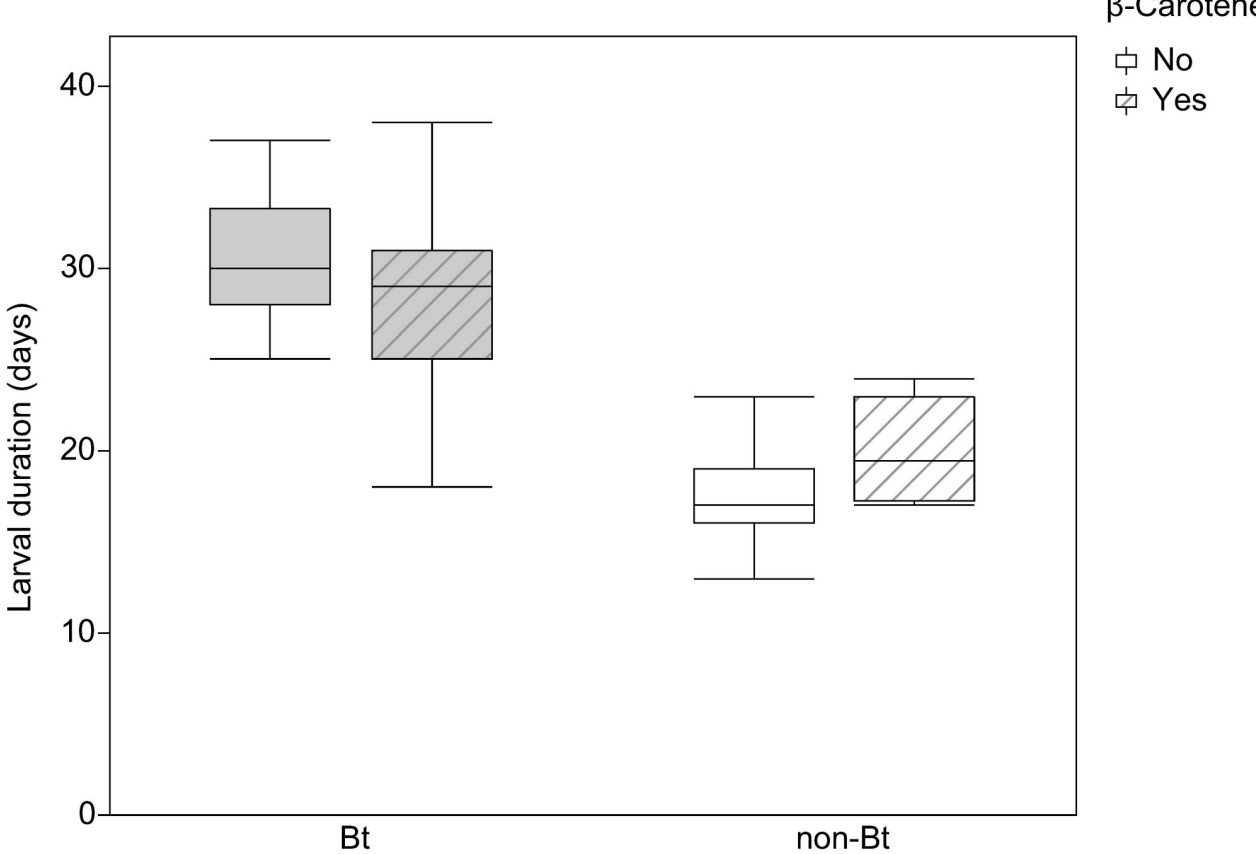

**Fig 2. Development of *Helicoverpa armigera* F1 larvae.** Effect of the addition of vitamin A (β-carotene: 0 or 0.6% in weight) to non-Bt or Bt diets on F1 larval duration (in days) of *Helicoverpa armigera*. Bt (grey boxplot) indicates diets prepared with lyophilized leaves of Bt maize while non-Bt (white boxplot) indicates diets prepared with lyophilized leaves of non-Bt maize. Lined boxplots represent the addition of β-carotene to the diets while plots without lines represent diets without β-carotene. Statistical test: Two-way ANOVA ($P < 0.005$), means were compared by the LS Means Student's test.

had a positive effect ($\chi^2 = 8.195$, fd = 1, $P_{\chi2} = 0.0042$). The presence of β-carotene ($\chi^2 = 0.2862$, fd = 1, $P_{\chi2} = 0.5962$) and the Bt–β-carotene interaction ($\chi^2 = 0.758499$, fd = 1, $P_{\chi2} = 0.3838$) had no effects on mating percentage.

Fertility was calculated as the number of fertile eggs from mated females. Assessment of fertility using ANOVA revealed no differences in the number of eggs laid based on larval diet type ($F_{23, 3} = 0.9171$; $P = 0.4505$) (Table 2).

## F2 Development and mortality

No significant differences were found in the duration of larval development ($F_{617, 1} = 0.6205$; $P = 0.4312$) or weight of the resulting pupae ($F_{617, 1} = 0.2510$; $P = 0.6166$) due to the sex factor.

The diet ingested by both the F2 and F1 larvae affected the duration of development. Analysis of the F2 dietary components demonstrated that the Bt toxin was the only component that influenced F2 larval development (Fig 5 and Table 3).

F2 pupal weight was affected by the diet consumed by the experimental F2 larvae but not the diet consumed by the parental F1 larvae. (Table 3 and Fig 6).

ANOVA of the dietary components indicated that pupal weight decreased in the presence of the Bt toxin or β-carotene.

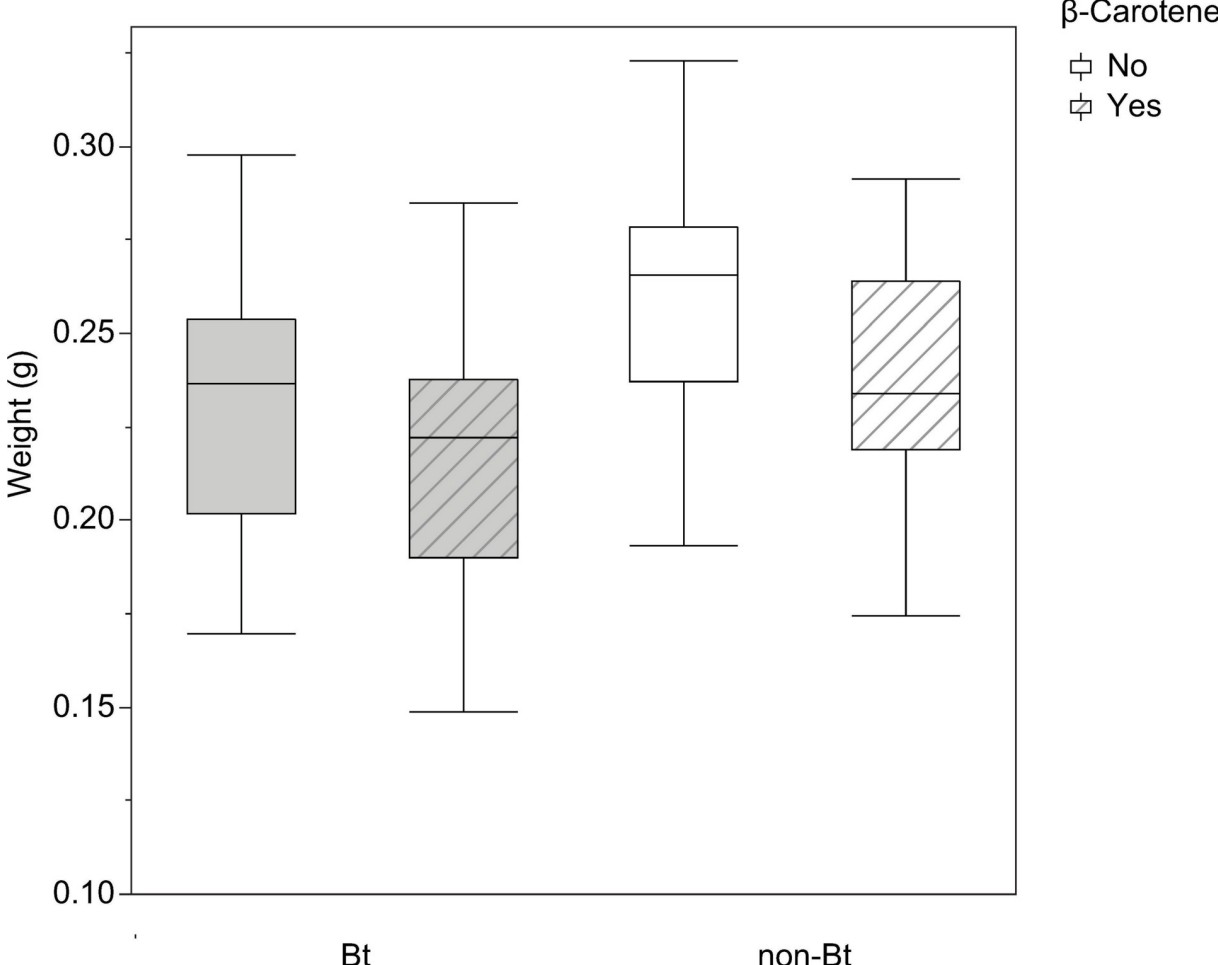

**Fig 3. Weight of *Helicoverpa armigera* F1 pupae.** Effect of the addition of vitamin A (β-carotene: 0 or 0.6% in weight) to non-Bt or Bt diets on the weight of the resulting F1 pupae of *Helicoverpa armigera*. Bt (grey boxplot) indicates diets prepared with lyophilized leaves of Bt maize while non-Bt (white boxplot) indicates diets prepared with lyophilized leaves of non-Bt maize. Lined boxplots represent the addition of β-carotene to the diets while plots without lines represent diets without β-carotene. Statistical test: Two-way ANOVA ($P < 0.005$), means were compared by the LS Means Student's test.

The diet of the parental F1 larvae affected the mortality of the F2 larval progeny ($\chi^2$ = 38.841; $P_{\chi2} < 0.0001$), as did the diet of these experimental larvae ($\chi^2$ = 158.37; $P_{\chi2} < 0.0001$) (Fig 7). Mortality was also affected by the interaction between the F1 larval diet and the experimental F2 diet ($\chi^2$ = 29.63; $P_{\chi2} < 0.0001$). Furthermore, mortality was increased in the F2

**Table 1. Mortality of the *Helicoverpa armigera* F1 larvae.**

| Diet | Mortality (%) |
|---|---|
| Bt | 7 |
| Bt+β | 23 |
| non-Bt | 16 |
| non-Bt+β | 11 |

Percentage mortality of *Helicoverpa armigera* F1 larvae fed on Bt or non-Bt diets with or without β-carotene supplementation during all stages.

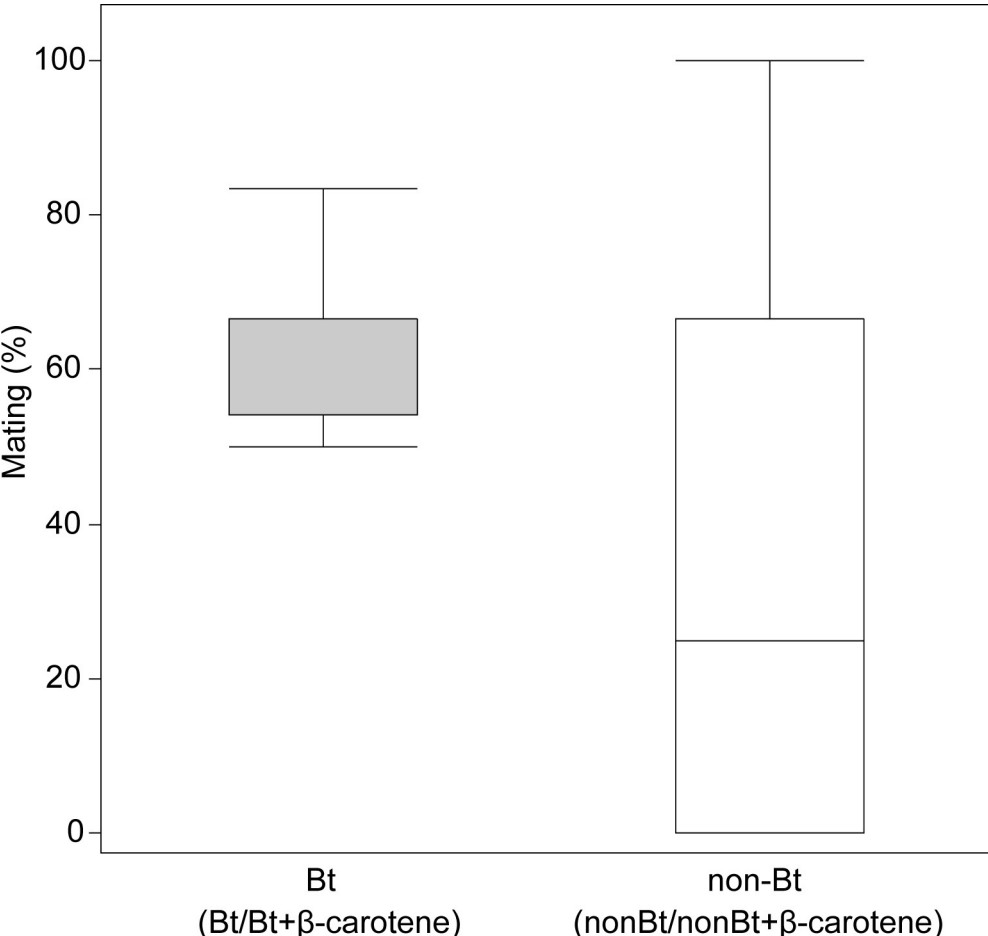

**Fig 4. Mating of the *Helicoverpa armigera* F1 adults.** Mating percentage of the *Helicoverpa armigera* F1 adults of larvae fed Bt or non-Bt diets with or without β-carotene supplementation during all stages. Bt (grey boxplot) indicates diets prepared with lyophilized leaves of Bt maize; non-Bt (white boxplot) indicates diets prepared with lyophilized leaves of non-Bt maize. As the presence of β-carotene had no effects on mating percentage, column Bt includes Bt and Bt+β-carotene results and column non-Bt includes non-Bt and non-Bt+β-carotene results.

larvae that ingested the Bt toxin in comparison to those that did not. The mortality of the F2 larvae descending from F1 larvae fed on Bt diets decreased in both F2 larvae that were fed on Bt diets and non-Bt diets (Fig 7). Analysis of dietary components indicated that only the presence of the Bt toxin in the diet affected the resulting mortality ($\chi^2$ = 140.26; $P_{\chi 2}$ < 0.0001)

**Table 2. Mating and fertility of *Helicoverpa armigera* F1 females.**

| DIET | % Mated females ($\overline{X}$ + se) | Fertile eggs/mated female ($\overline{X}$ + se) |
|---|---|---|
| Bt | 61.1 ± 10.24 | 459.9 ± 178 |
| Bt+β | 58.3 ± 8.33 | 296.8 ± 76.8 |
| non-Bt | 30.6 ± 13.89 | 508.11 ± 91.66 |
| non-Bt+β | 41.7 ± 19.12 | 373.63.8 ± 60.38 |

Percentage of mated *Helicoverpa armigera* F1 females of larvae fed Bt or non-Bt diets with or without β-carotene supplementation and number of fertile eggs of the mated F1 females.

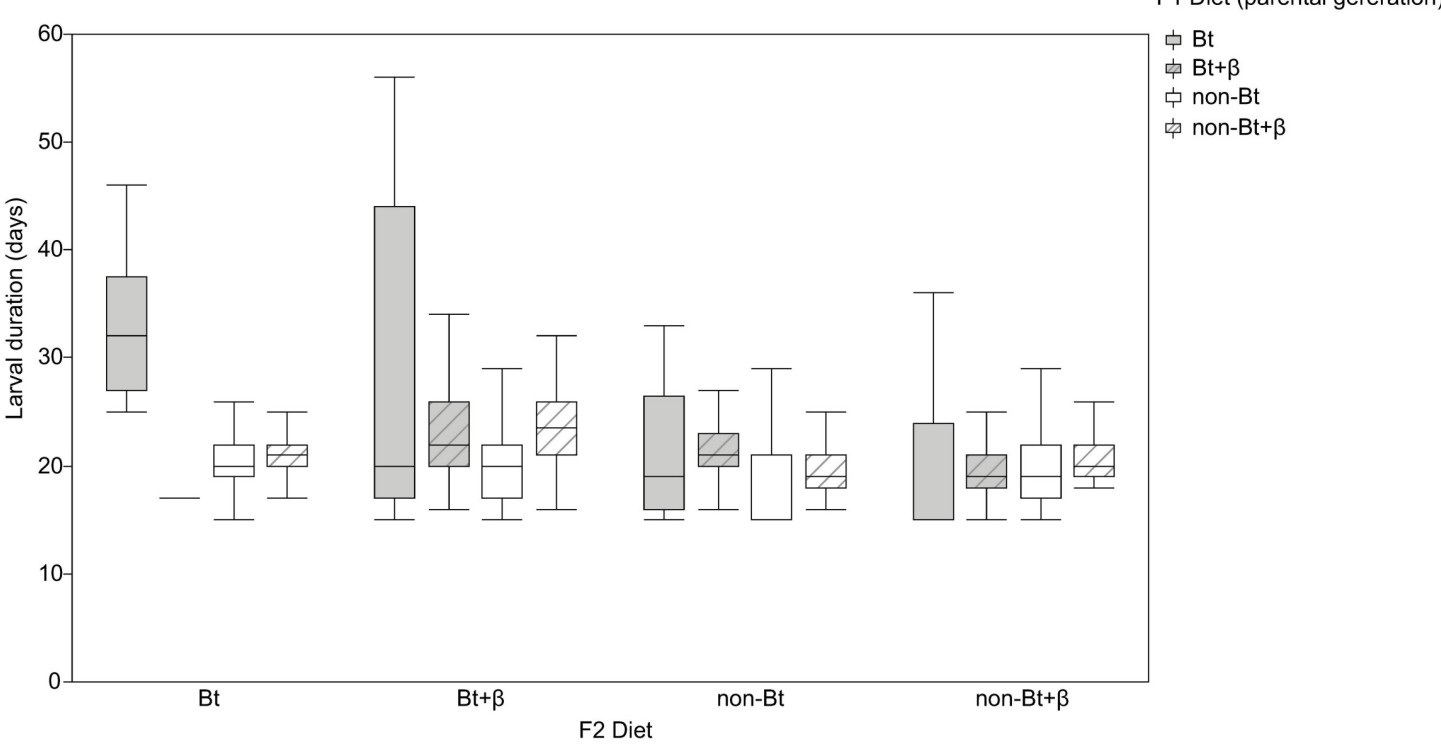

**Fig 5. Development of *Helicoverpa armigera* F2 larvae.** Larval duration (in days) of *Helicoverpa armigera* F2 larvae fed Bt or non-Bt diets with or without β-carotene supplementation coming from F1 larvae fed 1 of the 4 different diets. F2 diet indicates the diet of the experimental larvae; F1 diet indicates the diet of the parental larvae. Data were analyzed by a two-way ANOVA ($P < 0.005$); means were compared by LSD means.

(Fig 7). β-carotene ($\chi2 = 0.20488$; $P_{\chi2} < 0.6508$), and the interaction ($\chi2 = 1.08321$; $P_{\chi2} < 0.298$) did not affect the mortality.

## Discussion

Numerous studies on phytophagous species have analyzed life-history traits that are affected by the nutritional quality of the parents and the transgenerational effects thereof [25]. Parental and maternal effects have been detected in response to various biotic environmental factors, such as the quality and quantity of the host plant and the presence of pathogens. Previous

**Table 3. Statistics of the two-way ANOVA of larval development and pupal weight of *Helicoverpa armigera* F2.**

|  | Larval development | | | Pupal weight | | |
|---|---|---|---|---|---|---|
|  | F | fd | P | F | fd | P |
| Diet F1 | 22.55 | 632; 3 | < 0.0001 | 1.0380 | 629;3 | 0.3752 |
| Diet F2 | 12.97 | 632;3 | < 0.0001 | 180.82 | 629;3 | < 0.0001 |
| Diet F1 * Diet F2 | 10.2329 | 632;9 | < 0.0001 | 0.9307 | 629;3 | 0.4976 |
| Components F2 diet |  |  |  |  |  |  |
| Bt | 23.2974 | 632;1 | < 0.0001 | 600.924 | 629;1 | 0.0001 |
| β-carotene | 0.0421 | 632;1 | 0.8374 | 7.55 | 629;1 | 0.0062 |

Statistics (F, degrees of freedom, and *P*) were obtained using two-way ANOVA to analyze the influence of parental F1 larval diet and experimental F2 larval diet on F2 larval development and pupal weight. The last three rows show statistics from the two-way ANOVA of the F2 dietary components.

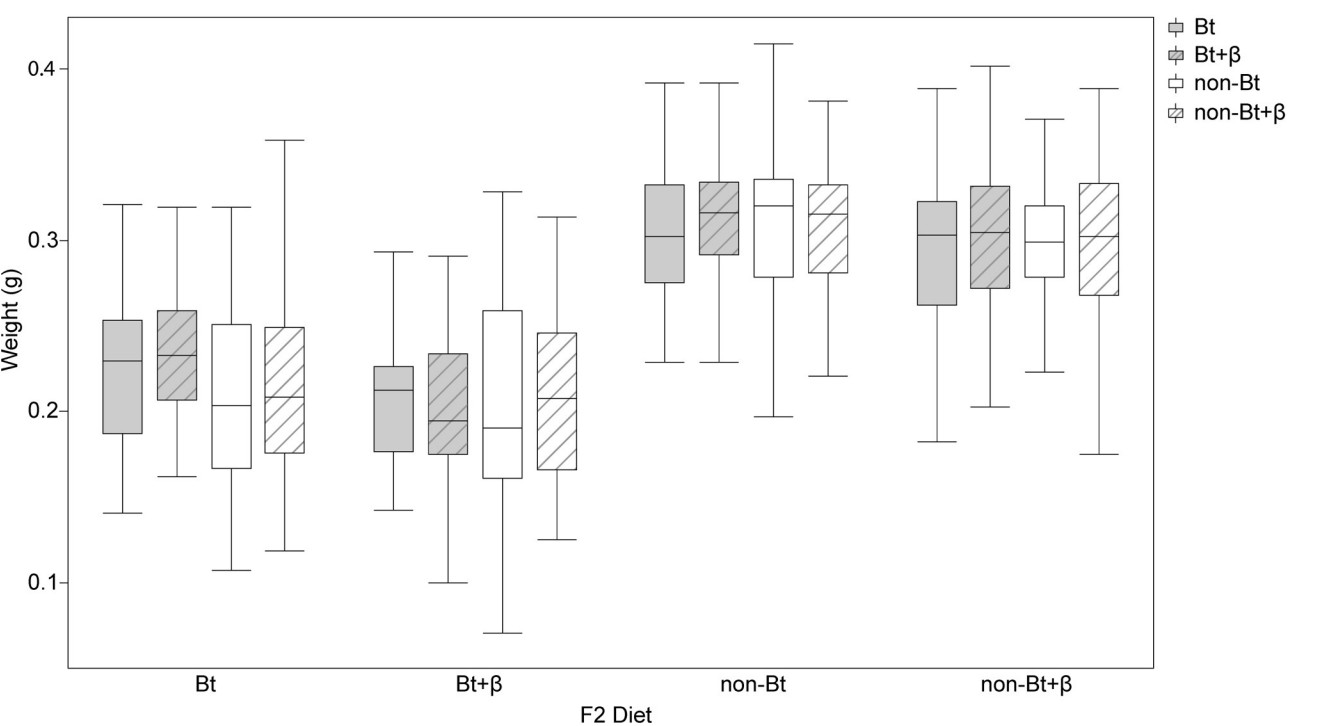

**Fig 6. Weight of the *Helicoverpa armigera* F2 pupae.** Weight of the *Helicoverpa armigera* pupae of F2 larvae fed Bt or non-Bt diets with or without β-carotene, from parental F1 larvae also fed 1 of the 4 experimental diets. F2 diet indicates the diet received by the experimental larvae while the F1 diet indicates the diet received by the parental larvae. Data were analyzed by two-way ANOVA ($P < 0.005$), and means were compared by an LSD test.

research has assessed life traits such as mortality, developmental duration, pupal mass, mating success, fertility, and offspring development and mortality [18, 25–27]. However, the effect of feeding on Bt crop on phytophagous life-history traits and the transgenerational effects thereof have received far less attention, perhaps due to the difficulty of collecting Bt-resistant larvae from Bt crops. To study these transgenerational effects, Guo *et al.* [28] collected *H. zea* larvae from seed blend refuges cops (RIB: seed blend refuges), sown with a mixture of Bt and non-Bt seeds, where the supposed that surviving larvae feed on sublethal doses of the toxin. Alternatively, Paula *et al.* [26] studied the intergenerational transfer of Bt proteins by feeding adults of non-target species on nectar with sublethal amounts of Bt toxin and by feeding larvae on leaves immersed in the Bt toxin. In the present study, neonate *H. armigera* larvae were fed, for 2 successive generations, on diet with a known concentration of Bt toxin [22] (a low-quality diet resulting in a sublethal dose of the Bt toxin) [29], or non-Bt leaves (a high-quality diet). The diets were or no supplemented with vitamin A. Using this model, the parental and transgenerational effects of diet quality on larval development, pupal weight, mating success, and female fertility were determined. The results confirmed that feeding on a Bt toxin diet throughout the larval instar prolongs development, producing lighter pupae from the larvae that survive. These findings have also been observed in a previous work on the same species, with a similar procedure (larvae fed on lyophilized Bt diet) [8] and on other related species, such as *H. zea* [28] but with larvae from field conditions, fed on sublethal amounts of the toxin. The parental effects of environmental nutritional factors on mating success have been studied scarcely; in this sense, Moreau *et al.* [25] related mating success with the quality of the parental diet. However, the present study demonstrated that *H. armigera* larvae that are fed on

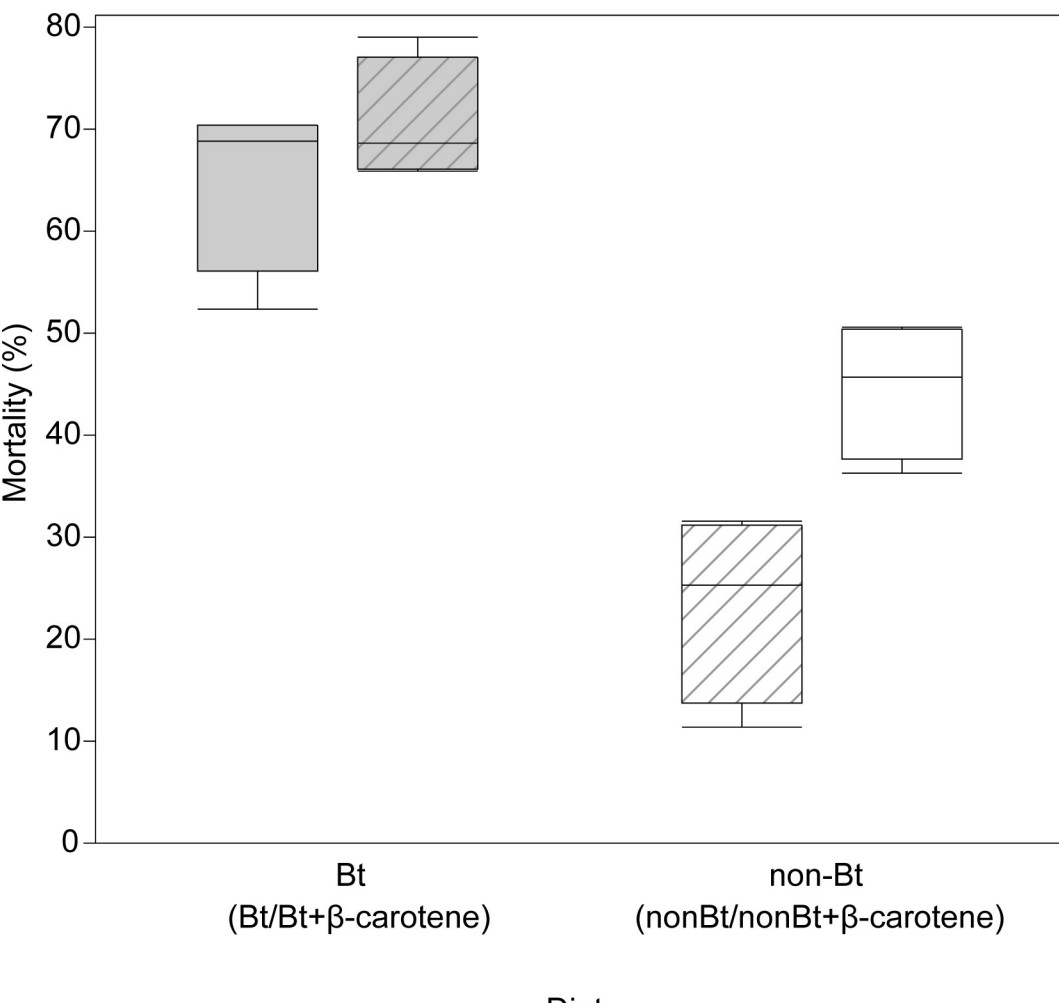

**Fig 7. Mortality of *Helicoverpa armigera* F2 larvae.** Percentage of the mortality of F2 larvae fed on Bt or non-Bt diets. The grey boxplots indicate F2 diets with Bt toxin; white boxplots indicate non-Bt diets. The lined grey boxplot indicates F2 larvae fed on Bt diet descending from F1 larvae fed on non-Bt diet; the lined white boxplot indicates F2 larvae fed on non-Bt diet descending from F1 larvae fed on Bt diet. As the presence of β-carotene had no effects on mortality percentage, columns Bt include Bt and Bt+β-carotene results and columns non-Bt include non-Bt and non-Bt+β-carotene results.

low-quality Bt diet produce lighter pupae and adults with greater mating success, while larvae that are fed a high-quality non-Bt diet produce heavier pupae and adults with lower mating success. Moreover, although Rossiter *et al.* [18] indicated that pupal weight is strongly correlated with fecundity in many Lepidoptera, this study shows that the fertility of *H. armigera* mated females is not affected by pupal weight. This was also observed in the fertility studies performed by Guo *et al.* [28, 30] on *H. zea*, and by Paula *et al.* [26] on *Chlosyne lacinia* (Geyer). Mating success and fertility could be two crucial factors that affect the performance and growth of *H. armigera* populations in Bt crops. According to Cahenzli and Erhardt [17], parental effects are one of the most important influences on the progeny phenotype. Regarding transgenerational effects, the results of this study indicate that the duration of larval development is longer when the parental larvae are fed a Bt diet instead of a non-Bt diet. Similar findings were obtained by Rossiter *et al.* [18], who fed *Lymantria dispar* diets of low nutritional quality and high phenolic content. The parental effect was also clearly seen in the lower

mortality of F2 individuals whose parents were fed a Bt diet instead of a non-Bt diet. This decrease in larval mortality could be considered an adaptive effect to the toxin or the low-quality diet, as noted by Quezada-Garcia *et al* [27] in *Chroristoneura fumiferana*. This adaptive effect may favor the development of resistance to Bt, which has been observed in *H. armigera* [14, 15], as well as *H. zea* [28].

Of importance, the addition of β-carotene to the diet was not beneficial for the pest in terms of larval development. On the contrary, it produced a reduction in pupal weight and prolonged developmental duration, confirming the conclusions of Lopez *et al.* [8], that the addition of vitamins to Bt crops should not be an advantage for the pests of these crops.

Maternal and paternal effects due to environmental factors on offspring are complex phenomena that have been studied from various viewpoints, such as the induction of diapause, nutritional or reproductive responses, causes of certain pest outbreaks, immune responses to pathogens, and acclimatization to host plants. Transgenic crops should, therefore, be considered from the perspective of the consequences involving the pests that feed on them. *H. armigera* is highly polyphagous that feeds on various crops from different coexisting botanical families in the region of the present study. Therefore, it would be interesting to study the effects of parental host crop type on the mating success and mortality of the subsequent generation. Similarly, it would be interesting to study the mating and ovipositor preferences of females developed in transgenic maize, particularly if they prefer to mate with males of the same crop and if, once mated, they preferred to oviposit in the transgenic maize, as described by Hopkins' host selection principle (HHSP), or by the chemical legacy hypothesis [31]. All of these factors play a part in determining the adaptability of *H. armigera* to Bt maize, thereby facilitating the development of resistance, which has already been detected in this species.

## Supporting information

**S1 File. The file contains data set needed to draw Figs 2–7 and Tables 1–3 Parental effects of Bt toxin and vitamin A on *Helicoverpa armigera*.**
(PDF)

## Acknowledgments

The authors thank Teresa Estela her administrative support and Joan Safont for his invaluable help in the general rearing of insects. Thanks to Professor Ramon Albajes for his unconditional support.

## Author Contributions

**Conceptualization:** Carmen López, Matilde Eizaguirre.

**Data curation:** Carmen López, Pilar Muñoz, Matilde Eizaguirre.

**Formal analysis:** Carmen López, Alejandro Juárez-Escario, Matilde Eizaguirre.

**Investigation:** Carmen López, Matilde Eizaguirre.

**Methodology:** Carmen López, Pilar Muñoz, Matilde Eizaguirre.

**Resources:** Carmen López, Pilar Muñoz, Matilde Eizaguirre.

**Visualization:** Carmen López, Alejandro Juárez-Escario, Matilde Eizaguirre.

**Writing – original draft:** Carmen López, Daniela Zanga, Matilde Eizaguirre.

**Writing – review & editing:** Carmen López, Daniela Zanga, Alejandro Juárez-Escario, Matilde Eizaguirre.

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
