## [Decision Letter · Decision Letter 0]

6 Apr 2022

PONE-D-22-04499Parental effects of Bt toxin and vitamin A on Helicoverpa armigeraPLOS ONE

Dear Dr. Malak

Thank you for submitting your manuscript to PLOS ONE. After careful consideration, we feel that it has merit but does not fully meet PLOS ONE’s publication criteria as it currently stands. Therefore, we invite you to submit a revised version of the manuscript that addresses the points raised during the review process.

Your manuscript has now been reviewed, and the reviewers' comments are appended below. You will see that, while they find your work of interest, they have raised some points that need to be addressed before we can make a decision on publication.

We look forward to receiving your revised manuscript.

Kind regards,

Patrizia Falabella

Academic Editor

PLOS ONE

Journal Requirements:

[We are grateful to Ministerio de Ciencia e Innovación (Spanish government) for financial support. The authors thank Teresa Estela her administrative support and Joan Safont for his invaluable help in the general rearing of insects. Thanks to Professor Ramon Albajes for his unconditional support.]

 [The funders had no role in study design, data collection and analysis, decision to publish, or preparation of the manuscript.]

Reviewers' comments:

Reviewer's Responses to Questions

**Comments to the Author**

1. Is the manuscript technically sound, and do the data support the conclusions?

Reviewer #1: Yes

Reviewer #2: Yes

2. Has the statistical analysis been performed appropriately and rigorously? 

Reviewer #1: Yes

Reviewer #2: Yes

3. Have the authors made all data underlying the findings in their manuscript fully available?

Reviewer #1: Yes

Reviewer #2: Yes

4. Is the manuscript presented in an intelligible fashion and written in standard English?

Reviewer #1: No

Reviewer #2: Yes

5. Review Comments to the Author

Reviewer #1: Comments:

This paper by López et al. “Parental effects of Bt toxin and vitamin A on Helicoverpa armigera” has researched the effect of vitamin A (β-carotene) and Bt protein on H. armigera larvae and their offspring. The work could be helpful to better understand the transgenerational effect of vitamin and Bt protein, and the potential resistance of H. armigera to Bt crops. In general, the experimental design of the paper is good and the data provided consequent, the information amassed is interesting. However, the manuscript is not acceptable before some experiment should be added. Meanwhile, the language quality of the MS should be improved. I hope my comments could help this interesting work

Introduction

The paragraphs of the Introduction is much too long. And there are too many long sentences, many of them are hard to understand.

Materials and methods

Insect rearing

“H. armigera larvae were collected in maize or alfalfa crops in Lleida”. Host adaptability of insect has been found to be dependent on many factors, the authors need to be clear that the larvae were collected from maize? or alfalfa crops?

Diets

“……, a 25% of maize flour was replaced by lyophilized Bt or non-Bt maize leaves, adding or not 0.6 % of β-Carotene”. The author needs to explain that why diets and 25% maize folur, 0.6% β-Carotene. Is there any literature on this? What was the dosage of Cry1Ab protein in 25% of Bt maize flour?

Experiments

Experiments with the F1 larvae and adults

“Larval mortality, duration of larval development and pupal weight were recorded”. Were the insect observed and recorded the mortality daily? What is the standard of death? Why were there 6 females and 5 males in the mating cage?

Please clearly state the number of insect for each experiment.

Statistical analysis

“using a generalized linear model with a binomial distribution” corrected to “using a generalized linear model (GLM) with a binomial distribution”. I recommend that you use this symbol instead of the common abbreviations throughout the manuscript.

Results

F1 mating and fertility

“……but neither β-carotene nor the interaction Bt-β-carotene had any effect”, the result of this sentence was not show in Fig. 4.

F2 Development and Mortality

Can the authors clarify why they did not research on the β-carotene to F2 development and mortality? Figure note should be added in Fig. 7.

Discussion

The paragraphs of the Discussion is much too long

“However, the life-history-traits affected by phytophagous feeding……”, please cite literature references.

“……in previous work with the same species [8] and with other nearby species such as H. zea [22]”, the authors should be discussed the differences between references and this research.

“Maternal o paternal effects due to……” corrected to “Maternal or paternal effects due to……”

Reviewer #2: Reviewer Comments for PONE-D-22-04499

The manuscript entitled “Parental effects of Bt toxin and vitamin A on Helicoverpa armigera” by

Carmen López, Daniela Zanga, Alejandro Juárez-Escario, Pilar Muñoz, Matilde Eizaguirre provides advances in the existing knowledge on Bt toxins for crop protection.

Introduction

developed instead of development

Add more reference against each already mentioned “H. armigera is a serious cosmopolitan insect pest whose larvae are polyphagous [9] and have a high capacity to develop resistance to insecticides [10] and to the Bt toxin [11].

Rephrase “that feed on maize mainly on the silks (styles) among many other crops such as cotton or alfalfa.”

(to the Bt toxin) and although part of the larvae feed on transgenic maize diet, another considerable part survives and pupates.

were collected from maize or alfalfa crops in Lleida

16:8 light: dark (h) photoperiod

relative humidity (>60%)

the pupae were placed in the adult-emerging conditions at 25ºC

Rephrase this “(cylindrical methacrylate tubes of 25 cm Ø and 40 cm high, closed at the base with an expanded polystyrene lid and at the top with a white gauze) with a drinker with honey solution and a cotton ball for egg laying.”

80⁰C until use.

diet made with non-Bt leaves (non-Bt),

rewrite in scientific way (5.5 cm ø x 3cm high) and use X from symbols not from alphabets

correct the space between degree and Celsius throughout the text of manuscript (e.g., 25 º C)

Better to use symbolic multiplication instead of others……for (6 cages* 6 females/cage) and 30 males (6 cages* 5 males/cage) for each condition. & ANOVA was used (the factors were, diet * sex)

Maintain space with * for two-way (parents*diet) ANOVA on development duration

Maintain space (Bt * β-carotene).

Space between fig and number (e.g., Fig1 = Fig 1)

Results

Percent mortality of the Helicoverpa

Discussion

Please update with latest references

Overall, please carefully crosscheck the reference mentioned in the text and list.

6. PLOS authors have the option to publish the peer review history of their article (what does this mean?). If published, this will include your full peer review and any attached files.

Reviewer #1: No

Reviewer #2: **Yes: **Abid Ali

---

## [Author Response · Author response to Decision Letter 0]

19 May 2022

PONE-D-22-04499

Parental effects of Bt toxin and vitamin A on Helicoverpa armigera

PLOS ONE

Patrizia Falabella Academic Editor and Reviewer's Questions

PLOS ONE

Journal Requirements:

When you resubmit, please ensure that you provide the correct grant numbers for the awards you received for your study in the ‘Funding Information’ section. Now Corrected

[We are grateful to Ministerio de Ciencia e Innovación (Spanish government) for financial support. The authors thank Teresa Estela her administrative support and Joan Safont for his invaluable help in the general rearing of insects. Thanks to Professor Ramon Albajes for his unconditional support.]

 [The funders had no role in study design, data collection and analysis, decision to publish, or preparation of the manuscript.]

Please include your amended statements within your cover letter; we will change the online submission form on your behalf. Done L 394

4. Please include your full ethics statement in the ‘Methods’ section of your manuscript file. In your statement, please include the full name of the IRB or ethics committee who approved or waived your study, as well as whether or not you obtained informed written or verbal consent. If consent was waived for your study, please include this information in your statement as well. Done, Line 108

Reviewers' comments:

Reviewer's Responses to Questions

Comments to the Author

1. Is the manuscript technically sound, and do the data support the conclusions?

Reviewer #1: Yes

Reviewer #2: Yes

2. Has the statistical analysis been performed appropriately and rigorously? 

Reviewer #1: Yes

Reviewer #2: Yes

3. Have the authors made all data underlying the findings in their manuscript fully available?

Reviewer #1: Yes

Reviewer #2: Yes

4. Is the manuscript presented in an intelligible fashion and written in standard English?

Reviewer #1: No The document has now been revised by an official English Editing services. We have the certificate showing it.

Reviewer #2: Yes

5. Review Comments to the Author

Reviewer #1: Comments:

This paper by López et al. “Parental effects of Bt toxin and vitamin A on Helicoverpa armigera” has researched the effect of vitamin A (β-carotene) and Bt protein on H. armigera larvae and their offspring. The work could be helpful to better understand the transgenerational effect of vitamin and Bt protein, and the potential resistance of H. armigera to Bt crops. In general, the experimental design of the paper is good and the data provided consequent, the information amassed is interesting. However, the manuscript is not acceptable before some experiment should be added. Meanwhile, the language quality of the MS should be improved. I hope my comments could help this interesting work

Introduction

The paragraphs of the Introduction is much too long. And there are too many long sentences, many of them are hard to understand. Done. The document has now been revised by an official English Editing services

Materials and methods

Insect rearing

“H. armigera larvae were collected in maize or alfalfa crops in Lleida”. Host adaptability of insect has been found to be dependent on many factors, the authors need to be clear that the larvae were collected from maize? or alfalfa crops?

The larvae of all experiments were collected only in alfalfa fields. Now this point has been corrected in M&M Line 108

Diets

“……, a 25% of maize flour was replaced by lyophilized Bt or non-Bt maize leaves, adding or not 0.6 % of β-Carotene”. The author needs to explain that why diets and 25% maize folur, 0.6% β-Carotene. Is there any literature on this? What was the dosage of Cry1Ab protein in 25% of Bt maize flour? Done. We have rewritten lines 129-136 adding the data and necessary references

Experiments

Experiments with the F1 larvae and adults

“Larval mortality, duration of larval development and pupal weight were recorded”. Were the insect observed and recorded the mortality daily? Yes, daily, line 149.

 What is the standard of death? We consider the standard mortality, the mortality on the non-Bt diet

Why were there 6 females and 5 males in the mating cage? In the M&M, Insect General Rearing we have added the explanation with the reference of why we used 6 females and 5 males per cage, lines 114-115

Please clearly state the number of insect for each experiment. Each experiment indicates the number on insects tested. Lines 145, 171

Statistical analysis

“using a generalized linear model with a binomial distribution” corrected to “using a generalized linear model (GLM) with a binomial distribution”. I recommend that you use this symbol instead of the common abbreviations throughout the manuscript. Done

Results

F1 mating and fertility

“……but neither β-carotene nor the interaction Bt-β-carotene had any effect”, the result of this sentence was not show in Fig. 4. We apologized for the unclear presentation of these results. Now, we have added the statistics of the effect of β-carotene in lines 244-246, we have also modified the X-axis the Figure 4 and the Figure 4 caption (lines 251-253) to explain in a better way.

F2 Development and Mortality

Can the authors clarify why they did not research on the β-carotene to F2 development and mortality? Figure note should be added in Fig. 7. Again, we apologized for the unclear presentation of these results. Now, we have added the statistics of the effect of β-carotene in lines 298-299, we have also modified the X-axis in the Figure 7, and the Figure 7 caption (lines 304-306) to explain in a better way.

Discussion

The paragraphs of the Discussion is much too long

Done. The document has been revised.

“However, the life-history-traits affected by phytophagous feeding……”, please cite literature references. 

The literature references appear in line 314

“……in previous work with the same species [8] and with other nearby species such as H. zea [22]”, the authors should be discussed the differences between references and this research. Done, lines 329-332

“Maternal o paternal effects due to……” corrected to “Maternal or paternal effects due to……” Changed in the new version, line 360

Reviewer #2: Reviewer Comments for PONE-D-22-04499

The manuscript entitled “Parental effects of Bt toxin and vitamin A on Helicoverpa armigera” by

Carmen López, Daniela Zanga, Alejandro Juárez-Escario, Pilar Muñoz, Matilde Eizaguirre provides advances in the existing knowledge on Bt toxins for crop protection.

Introduction

developed instead of development Changed in the new version

Add more reference against each already mentioned “H. armigera is a serious cosmopolitan insect pest whose larvae are polyphagous [9] and have a high capacity to develop resistance to insecticides [10] and to the Bt toxin [11]. Done, added more references. L74-77

Rephrase “that feed on maize mainly on the silks (styles) among many other crops such as cotton or alfalfa.” Eliminated in the new version 

(to the Bt toxin) and although part of the larvae feed on transgenic maize diet, another considerable part survives and pupates. Rewritten in the new version 

were collected from maize or alfalfa crops in Lleida Changed, line 108

16:8 light: dark (h) photoperiod Corrected

relative humidity (>60%) Corrected

the pupae were placed in the adult-emerging conditions at 25ºC

Rephrase this “(cylindrical methacrylate tubes of 25 cm Ø and 40 cm high, closed at the base with an expanded polystyrene lid and at the top with a white gauze) with a drinker with honey solution and a cotton ball for egg laying.” All M&M have been rewritten 

80⁰C until use. -80 º C Corrected

diet made with non-Bt leaves (non-Bt), M&M have been rewritten

rewrite in scientific way (5.5 cm ø x 3cm high) and use X from symbols not from alphabets Done

correct the space between degree and Celsius throughout the text of manuscript (e.g., 25 º C) Done

Better to use symbolic multiplication instead of others……for (6 cages* 6 females/cage) and 30 males (6 cages* 5 males/cage) for each condition. & ANOVA was used (the factors were, diet * sex) Maintain space with * for two-way (parents*diet) ANOVA on development duration Done

Maintain space (Bt * β-carotene). Done

Space between fig and number (e.g., Fig1 = Fig 1) Done

Results

Percent mortality of the Helicoverpa Corrected

Discussion

Please update with latest references We have used the latest references we have found

Overall, please carefully crosscheck the reference mentioned in the text and list.

6. PLOS authors have the option to publish the peer review history of their article (what does this mean?). If published, this will include your full peer review and any attached files.

Do you want your identity to be public for this peer review? For information about this choice, including consent withdrawal, please see our Privacy Policy.

Reviewer #1: No

Reviewer #2: Yes: Abid Ali

---

## [Editor Report · Decision Letter 1]

25 May 2022

Parental effects of Bt toxin and vitamin A on Helicoverpa armigera

PONE-D-22-04499R1

Dear Dr. Eizaguirre,

We’re pleased to inform you that your manuscript has been judged scientifically suitable for publication and will be formally accepted for publication once it meets all outstanding technical requirements.

Kind regards,

Patrizia Falabella

Academic Editor

PLOS ONE
---

## [Editor Report · Acceptance letter]

23 Jun 2022

PONE-D-22-04499R1 

Parental effects of Bt toxin and vitamin A on *Helicoverpa armigera*

Dear Dr. Eizaguirre:

I'm pleased to inform you that your manuscript has been deemed suitable for publication in PLOS ONE. Congratulations! Your manuscript is now with our production department. 

Kind regards, 

on behalf of

Prof. Patrizia Falabella 

Academic Editor

PLOS ONE